# Matching the Budyko functions with the complementary evaporation relationship: consequences for the drying power of the air and the Priestley-Taylor coefficient

Jean-Paul Lhomme[1], Roger Moussa[2]

[1]IRD, UMR LISAH, 2 Place Viala, 34060 Montpellier, France
[2]INRA, UMR LISAH, 2 Place Viala, 34060 Montpellier, France

*Correspondence to*: Roger Moussa (moussa@supagro.inra.fr)

**Abstract.** The Budyko functions $B_1(\Phi_p)$ are dimensionless relationships relating the ratio $E/P$ (actual evaporation over precipitation) to the aridity index $\Phi_p = E_p/P$ (potential evaporation over precipitation). They are valid at catchment scale with $E_p$ generally defined by Penman's equation. The complementary evaporation (CE) relationship stipulates that a decreasing actual evaporation enhances potential evaporation through the drying power of the air which becomes higher. The Turc-Mezentsev function with its shape parameter $\lambda$, chosen as example among various Budyko functions, is matched with the CE relationship, implemented through a generalized form of the Advection-Aridity model. First, we show that there is a functional dependence between the Turc-Budyko curve and the drying power of the air. Then, we examine the case where potential evaporation is calculated by means of a Priestley-Taylor type equation ($E_0$) with a varying coefficient $\alpha_0$. Matching the CE relationship with the Budyko function leads to a new transcendental form of the Budyko function $B_1{}'(\Phi_0)$ linking $E/P$ to $\Phi_0 = E_0/P$. For the two functions $B_1(\Phi_p)$ and $B_1{}'(\Phi_0)$ to be equivalent, the Priestley-Taylor coefficient $\alpha_0$ should have a specified value as a function of the Turc-Mezentsev shape parameter and the aridity index. This functional relationship is specified and analysed.

## 1 Introduction

The Budyko curves are analytical formulations of the functional dependence of actual evaporation $E$ on moisture availability, represented by precipitation $P$, and atmospheric water demand, represented by potential evaporation $E_p$. They are valid on long timescales at catchment scale. More precisely, the Budyko functions relate the evaporation fraction $E/P$ to an aridity index defined as $\Phi_p = E_p/P$. Empirical formulations have been obtained by simple fitting to observed values (Turc, 1954; Budyko, 1974). Analytical derivations have also been developed (Mezentsev, 1955; Fu, 1981; Zhang et al., 2004; Yang et al., 2008). The Budyko relationships have been extensively used in the scientific literature up to now and interpreted with physical models (Gerrits et al., 2009) or thermodynamic approaches (Wang et al., 20015). For some of the formulations the shape of the curve is determined by a parameter linked to catchment characteristics such as vegetation, soil water storage

(Li et al., 2013; Yang et al., 2007) or catchment slope (Yang et al., 2014). The most representative functions $E/P = B(\Phi_p)$ are shown in Table 1 (see Lebecherel et al. (2013) for an historical overview) and one of them (Turc-Mezentsev) is represented in Fig. 1 for different values of its shape parameter. All the Budyko functions assume steady-state conditions, which means that all the water consumed by evaporation comes from the precipitation and that the change in catchment water storage is nil: $P-E = Q$ with $Q$ the total runoff. Consequently, the following conditions should be met: (i) $E = 0$ if $P = 0$, (ii) $E \leqslant P$ (water limit), (iii) $E \leqslant E_p$ (energy limit), (iv) $E \rightarrow E_p$ if $P \rightarrow +\infty$. These conditions define a physical domain where the Budyko curves are constrained (Fig. 1). It is interesting to note also that any Budyko function $B_1$ relating $E/P$ to $\Phi_p$ can be transformed into a corresponding function $B_2$ relating $E/E_p$ to $\Phi_p^{-1} = P/E_p$ (Zhang et al., 2004; Yang et al., 2008). Indeed we have:

$$\frac{E}{E_p} = B_2\left(\Phi_p^{-1}\right) = \frac{E}{P}\frac{P}{E_p} = B_1(\Phi_p)\Phi_p^{-1} = \Phi_p^{-1}B_1\left(\frac{1}{\Phi_p^{-1}}\right). \tag{1}$$

Potential evaporation establishes an upper limit to the evaporation process in a given environment. It is generally given by a Penman-type equation (Lhomme, 1997a), which is the sum of two terms: a first term depending on the radiation load $R_n$ and a second term involving the drying power of the ambient atmosphere $E_a$:

$$E_P = \frac{\Delta}{\Delta+\gamma}R_n + \frac{\gamma}{\Delta+\gamma}E_a. \tag{2}$$

In Eq. (2) $\gamma$ is the psychrometric constant and $\Delta$ the slope of the saturated vapour pressure curve at air temperature. $E_a$ represents the capacity of the ambient air to extract water from the surface. It is an increasing function of the vapour pressure deficit of the air $D_a$ and of wind speed $u$ through a wind function $f(u)$: $E_a = f(u) D_a$. Contrary to precipitation, potential evaporation $E_p$ is not a forcing variable independent of the surface. $E_p$ is in fact coupled to $E$ by means of a functional relationship known as the complementary evaporation relationship (Bouchet, 1963), which stipulates that potential evaporation increases when actual evaporation decreases. This complementary behaviour is made through the drying power of the air $E_a$: a decreasing actual evaporation makes the ambient air drier, which enhances $E_a$ and thus potential evaporation. Eq. (2) takes into account this complementary behaviour through the drying power $E_a$, which adjusts itself to the conditions generated by the rate of actual evaporation. It is also the case, for instance, when $E_p$ is calculated as a function of pan evaporation.

In most of Budyko type functions encountered in the literature, potential evaporation $E_p$ is generally not defined with accuracy. Choudhury (1999, p. 100) noted that "varied methods were used to calculate $E_p$, and these methods can give substantially different results". Moreover, in the original framework and in some subsequent works (e.g. Choudhury, 1999; Donohue et al., 2007), net radiation alone is used as a good approximation of the energy available for evaporation. Many formulae, in fact, can be used to calculate the potential rate of evaporation, each one involving different weather variables and yielding different values. Some formulae are based upon temperature alone, others on temperature and radiation (Carmona et al., 2016). In the present study we examine the case where $E_p$ is estimated via a Priestley-Taylor type equation (Priestley and Taylor, 1978) with a variable coefficient $\alpha_0$:

$$E_0 = \alpha_0 \frac{\Delta}{\Delta + \gamma} R_n \ . \tag{3}$$

Here, soil heat flux is neglected on large timescale and the coefficient $\alpha_0$ (named "Priestley-Taylor" coefficient) has not the fixed value (1.26) mentioned in the original work of Priestley-Taylor. It is supposed to increase with climate aridity and could vary from around *1.25* up to *1.75* according to Shuttleworth (2012). This can be seen as a direct consequence of the

complementary evaporation relationship. Indeed, $\alpha_0$ is linked to $E_a$ by $\alpha_0 = 1 + (\gamma/\Delta)E_a/R_n$ (obtained by matching Eqs. 2 and 3), which shows that $\alpha_0$ increases when the drying power rises. Lhomme (1997b) made a thorough examination of the so defined coefficient $\alpha_0$ by means of a convective boundary layer model.

      In the present paper, the behaviour of the drying power of the air $E_a$ will be examined, together with its physical boundaries, in relation to the actual rate of evaporation predicted by the Budyko functions. We will also show that the

coefficient $\alpha_0$, which allows an estimate of potential evaporation through the Priestley-Taylor equation (Eq. 3), has a functional relationship with the shape parameter of the Budyko curve and the aridity index, this last point constituting our main objective. Once $\alpha_0$ is determined and thus potential evaporation $E_0$, actual evaporation can be estimated, either through the Budyko function or the CE relationship. The standpoint used in the study differs from various previous attempts undertaken in the literature to examine from different perspectives the links between Bouchet and Budyko relationships,

investigating their apparent contradictory behaviour (Szilagyi and Jozsa, 2009). For example, Zhang et al. (2004) established a parallel between the assumptions underlying Fu's equation and the complementary relationship. In a study by Yang et al. (2006) concerning numerous catchments in China, the consistency between Bouchet, Penman and Budyko hypotheses was theoretically and empirically explained. Lintner et al. (2015) examined the Budyko and complementary relationships using an idealized prototype representing the physics of large-scale land-atmosphere coupling in order to evaluate the

anthropogenic influences. Zhou et al. (2015) developed a complementary relationship for partial elasticities to generate Budyko functions, their relationship fundamentally differing from Bouchet's one. Carmona et al. (2016) proposed a power law to overcome a physical inconsistency of the Budyko curve in humid environments, this new scaling approach implicitly incorporating the complementary evaporation relationship.

      The paper is organized as follows. First, the basic equations used in the development are detailed: the choice of a

particular Budyko function is discussed and the complementary evaporation relationship, implemented through a generalised form of the Advection-Aridity model (Brutsaert and Stricker, 1979) is presented. Second, the feasible domain of the drying power of the air $E_a$ is examined, together with its correspondence in dimensionless form with actual evaporation, as predicted by the Budyko function. Third, the functional relationship linking the Priestley-Taylor coefficient $\alpha_0$ to the shape parameter of the Budyko function and the aridity index is inferred. In the following development, "complementary

evaporation" is abbreviated in CE.

## 2 Basic equations

Among the Budyko functions given in Table 1, one particular form is retained in our study: the one initially obtained by Turc (1954) and Mezentsev (1955) through empirical considerations and then analytically derived by Yang et al. (2008) through the resolution of a Pfaffian differential equation with particular boundary conditions. Three reasons guided this choice: (i) the function is one of the most commonly used; (ii) it involves a model parameter $\lambda$ which allows it to evolve within the Turc-Budyko physical domain; (iii) it has a notable simple mathematical property expressed as: $F(1/x) = F(x)/x$. This last property means that the same mathematical expression is valid for $B_1$ and $B_2$ (Eq. 1). The so-called Turc-Mezentsev function is expressed as:

$$\frac{E}{P} = B_1(\Phi_p) = \Phi_p \left[ 1 + (\Phi_p)^\lambda \right]^{-\frac{1}{\lambda}} = \left[ 1 + (\Phi_p)^{-\lambda} \right]^{-1/\lambda} . \tag{4}$$

It is written here with an exponent noted $\lambda$ instead of the $n$ generally used (Yang et al., 2009). The slope of the curve for $\Phi_p = 0$ is $1$. When the model parameter $\lambda$ increases from $0$ to $+\infty$, the curves grow from the x-axis (zero evaporation) to an upper limit (water and energy limits), as shown in Fig. 1. In other words, when $\lambda$ increases, actual evaporation gets closer to its maximum rate and when $\Phi_p$ tends to infinite $E/P$ tends to $1$. The intrinsic property of Eq. (4) allows it to be transformed into a similar equation with $E/E_p$ replacing $E/P$ and $\Phi_p^{-1}$ replacing $\Phi_p$ (see Figs. 2a, b):

$$\frac{E}{E_p} = B_2(\Phi_p^{-1}) = \Phi_p^{-1} \left[ 1 + (\Phi_p^{-1})^\lambda \right]^{-\frac{1}{\lambda}} = \left[ 1 + (\Phi_p^{-1})^{-\lambda} \right]^{-1/\lambda} . \tag{5}$$

Fu (1981) and Zhang et al. (2004) derived a very similar equation with a shape parameter $\omega$ (see Table 1) and Yang et al. (2008) established a simple linear relationship between the two parameters ($\omega = \lambda + 0.72$). In the rest of the paper, the development and calculations are made with the Turc-Mezentsev formulation. However, similar (but less straightforward) results can be obtained with the Fu-Zhang formulation (see the supplementary material S4).

The complementary evaporation (CE) relationship expresses that actual evaporation $E$ and potential evaporation $E_p$ are related in a complementary way following:

$$E_p + bE = (1 + b)E_w . \tag{6}$$

$E_w$ is the wet environment evaporation, which occurs when $E = E_p$ and $b \geqslant 1$ is a proportionality coefficient which accounts for the asymmetry of the relationship (Han et al., 2012): the increase in potential evaporation is generally higher than the reduction in actual evaporation. Various forms of the CE relationship exist in the literature (Xu et al., 2005; Brutsaert, 2015; Szilagyi et al., 2016) and the value of $b$ has been largely discussed (Kahler and Brutsaert, 2006; Pettijohn and Salvucci, 2009; Aminzadeh et al., 2015). In our analysis, the CE relationship is interpreted in the widely accepted framework of the Advection-Aridity model (Brutsaert and Stricker, 1979), where $b$ is assumed to be equal to 1, potential evaporation $E_p$ is calculated using Penman's equation (Eq. 2) and $E_w$ is expressed by the original Priestley-Taylor equation with a fixed value (1.26) of the coefficient $\alpha_w$:

$$E_w = \alpha_w \frac{\Delta}{\Delta + \gamma} R_n . \tag{7}$$

$E_w$ only depends on net radiation and air temperature through $\Delta$. The value of $\alpha_w$ has been the subject of discussion (Mallick et al., 2013): its analytical expression inferred from a land-atmosphere coupling model by Lintner et al. (2015) tends to prove that it could be lower than 1.26, in line with the in situ observations of Kahler and Brutsaert (2006). The value of 1.26, nevertheless, is kept in our numerical simulations, together with the value of 1 for $b$. All the algebraic calculations, however, will be performed with non-prescribed values of $b$ and $\alpha_w$, which allows other possible numerical simulations.

At this stage of the development it is important to make clear that two different Priestley-Taylor coefficients are defined in our analysis in relation to the CE relationship: one ($\alpha_w$) is used to define the wet environment evaporation $E_w$ and the other ($\alpha_0$) to calculate the potential evaporation $E_0$, which is a substitute for the "true" potential evaporation $E_p$ represented by Penman's equation (Eq. 2). Observational data confirm that the CE relationship generally holds on daily to annual timescales (Lintner et al., 2015). If the Budyko functions were initially derived and used on long timescales, they have been subsequently downscaled to the season or the month by some authors (Zhang et al., 2008; Du et al., 2016; Greve et al., 2016). This means that the matching between the two relationships is legitimate. $E_0$ (Eq. 3) being a substitute for $E_p$, it should also verify the CE relationship (Eq. 6), which implies that: $\alpha_w \leqslant \alpha_0 \leqslant (1+b)\alpha_w$.

As already said in the introduction, the complementarity between $E$ and $E_p$ is essentially made through the drying power of the air $E_a$: a decrease in regional actual evaporation, consecutive to a decrease in water availability, generates a drier air, which enhances $E_a$ and thus $E_p$. The behaviour of $E_a$ is examined in the next section.

**3 Feasible domain of the drying power of the air and correspondence with the evaporation rate**

As a consequence of land-atmosphere interactions expressed by the CE relationship, the drying power of the air $E_a$ is linked to the evaporation rate. Its feasible domain is examined hereafter by determining its bounding frontiers and its behaviour is assessed as a function of the evaporation rate. Inverting Eq. (2) and replacing its radiative term by $E_w$ (Eq. 7) yields:

$$E_a = \left(1 + \frac{\Delta}{\gamma}\right)\left(E_p - \frac{E_w}{\alpha_w}\right). \tag{8}$$

Taking into account the CE relationship (Eq. 6) and scaling by $E_p$ leads to:

$$\frac{E_a}{E_p} = \left(1 + \frac{\Delta}{\gamma}\right)\left[1 - \frac{1}{(1+b)\alpha_w}\left(1 + b\frac{E}{E_p}\right)\right]. \tag{9}$$

Inserting Eq. (5) into Eq. (9) gives:

$$\frac{E_a}{E_p} = D\left(\Phi_p^{-1}\right) = \left(1 + \frac{\Delta}{\gamma}\right)\left(1 - \frac{1}{(1+b)\alpha_w}\left\{1 + b\Phi_p^{-1}\left[1 + \left(\Phi_p^{-1}\right)^\lambda\right]^{-\frac{1}{\lambda}}\right\}\right). \tag{10}$$

This means that the ratio $E_a/E_p$ can be also expressed and drawn as a function of $\Phi_p^{-1}$ like the Budyko functions. Given that there is a water limit expressed by $0 < E < P$ and an energy limit expressed by $0 < E < E_p$, the function $E_a/E_p = D(\Phi_p^{-1})$ should meet the following three conditions:

(i) $E > 0$ implies that $E_a < E_{a,x}$ given by:

$$\frac{E_{a,x}}{E_P} = (1 + \frac{\Delta}{\gamma})\left[1 - \frac{1}{(1+b)\alpha_w}\right]. \tag{11}$$

(ii) $E < P$ implies that $E_a > E_{a,n1}$ given by:

$$\frac{E_{a,n1}}{E_p} = \left(1 + \frac{\Delta}{\gamma}\right)\left[1 - \frac{1}{(1+b)\alpha_w}\left(1 + b\frac{P}{E_P}\right)\right]. \tag{12}$$

(iii) $E < E_p$ implies that $E_a > E_{a,n2}$ given by:

$$\frac{E_{a,n2}}{E_p} = \left(1 + \frac{\Delta}{\gamma}\right)\left(1 - \frac{1}{\alpha_w}\right). \tag{13}$$

With $E_p$ as scaling parameter, the feasible domain of $E_a/E_p$ in the dimensionless space ($\Phi_p^{-1} = P/E_p$, $E_a/E_p$) is shown in Fig. 2c with $b=1$: when evaporation is nil, $E_a = E_{a,x}$ is maximum (upper boundary in Fig. 2c); when evaporation is maximal, $E_a$ is minimal (lower boundary in Fig. 2c). The maximum dimensionless difference $D^*$ between the upper boundary ($E_{a,x}/E_p$) and the lower boundary is obtained by subtracting Eq. (13) from Eq. (11):

$$D^* = \frac{b}{(1+b)\alpha_w}\left(1 + \frac{\Delta}{\gamma}\right). \tag{14}$$

There is a correspondence between the Budyko curves $E/P = B_1(\Phi_p)$ and $E/E_P = B_2(\Phi_p^{-1})$ drawn into Figs. 2a, b and the one of $E_a/E_p = D(\Phi_p^{-1})$ drawn in Fig. 2c. Figs. 2a, b, c show this correspondence for a particular case defined by $b = 1$, $\lambda = 1$ and *$\Delta = 110\ Pa\ °C^{-1}\ (T = 15°C)$*. When the Budyko curves reach their upper limit, i.e. in very evaporative environments, the corresponding curve $E_a/E_p$ reaches its lower limit. Conversely, when the Budyko curves reach their lower limit, i.e. the x-axis (no-evaporative environment), the corresponding $E_a/E_p$ curve reaches its upper limit.

It is interesting to note that the shape parameter $\lambda$ of the Turc-Mezentsev function has a clear graphical expression. Indeed, denoting by $d^*$ the maximum difference between the Turc-Mezentsev curve and its upper limit (Fig. 2a), this difference ($0 < d^* < 1$) obviously occurring for $\Phi_p = P/E_p = 1$, we have from Eq. (4):

$$d^* = 1 - 2^{-\frac{1}{\lambda}}, \tag{15}$$

which leads to:

$$\lambda = \frac{-ln2}{\ln(1-d^*)}. \tag{16}$$

When $d^*$ varies from *1* to *0*, the parameter $\lambda$ varies from *0* to $+\infty$. The value corresponding to $d^*$ in the graphical representation of $E_a/E_p = D(\Phi_p^{-1})$ (Fig. 2c) is the difference $\delta^*$ between the $E_a/E_p$ curve (Eq. 10) and its lower boundary (Eq. 13) for $\Phi_p^{-1} = P/E_p = 1$. It is given by

$$\delta^* = \left(1 + \frac{\Delta}{\gamma}\right)\frac{b}{(1+b)\alpha_w}\left(1 - 2^{-\frac{1}{\lambda}}\right) = D^*d^*. \tag{17}$$

This simple relationship shows that the dimensionless differences $d^*$ and $\delta^*$ vary simultaneously in the same direction with a proportionality coefficient equal to $D^*$, whose value is close to *1*. It is a direct consequence of the CE relationship. When $d^*$ decreases, i.e. the dimensionless evaporation rate ($E/P$ or $E/E_p$) increases, $\delta^*$ decreases, i.e. the drying power of the air $E_a$ decreases: the air becomes wetter (assuming a constant wind speed). In the next section, another consequence of the CE

relationship will be examined in relation to the value of the Priestley-Taylor coefficient $\alpha_0$ and its dependence on the rate of actual evaporation.

## 4 Linking the Priestley-Taylor coefficient to the Budyko functions

Using the CE relationship as a basis, this section examines the link existing between the Priestley-Taylor coefficient $\alpha_0$ defined by Eq. (3) and the Turc-Mezentsev shape parameter $\lambda$ (Eq. 4). Combining Eqs. (3), (6) and (7) potential evaporation can be written as:

$$E_p = (1 + b)\frac{\alpha_w}{\alpha_0}E_0 - bE \ . \tag{18}$$

Substituting $E_p$ in Eq. (4) by its value given by Eq. (18) and putting $\Phi_0 = E_0/P$ gives

$$\frac{E}{P} = \left[\frac{(1+b)\alpha_w}{\alpha_0}\Phi_0 - b\frac{E}{P}\right]\left\{1 + \left[\frac{(1+b)\alpha_w}{\alpha_0}\Phi_0 - b\frac{E}{P}\right]^\lambda\right\}^{-1/\lambda} \ . \tag{19}$$

Eq. (19) can be rewritten as:

$$\Phi_0 = B_1'^{-1}\left(\frac{E}{P}\right) = \frac{\alpha_0}{(1+b)\alpha_w}\left\{\left[\left(\frac{E}{P}\right)^{-\lambda} - 1\right]^{-1/\lambda} + b\frac{E}{P}\right\} \ . \tag{20}$$

Eq. (20) represents a transcendental form of the Turc-Mezentsev function (Eq. 4) issued from the complementary relationship and written with $\Phi_0 = E_0/P$ instead of $\Phi_p = E_p/P$. Calling $B_1'$ this new function $E/P = B_1'(\Phi_0)$, Eq. (20) represents in fact its inverse function $\Phi_0 = B_1'^{-1}(E/P)$. The function $E/P = B_1'(\Phi_0)$ has properties similar to the Turc-Mezentsev function (Eq. 4) (see the demonstrations in the supplementary materials S1): i) when $\Phi_0$ tends to zero, $B_1'(\Phi_0)$ tends to zero with a slope equal to $\alpha_w/\alpha_0$ ($\leqslant 1$); ii) when $\Phi_0$ tends to infinite, $E/P$ tends to $1$. A transcendental form of Eq. (5), called $B_2'$, can be obtained by expressing $E/E_0$ as a function of $\Phi_0^{-1} = P/E_0$:

$$\Phi_0^{-1} = B_2'^{-1}\left(\frac{E}{E_0}\right) = \left\{\left(\frac{E}{E_0}\right)^{-\lambda} - \left[\frac{(1+b)\alpha_w}{\alpha_0} - b\frac{E}{E_0}\right]^{-\lambda}\right\}^{-1/\lambda} \ . \tag{21}$$

Function $B_2'$ has the following properties at its limits (see the supplementary materials S2): i) when $\Phi_0^{-1}$ tends to zero, $B_2'(\Phi_0^{-1})$ tends to zero with a slope equal to 1; ii) when $\Phi_0^{-1}$ tends to infinite, $E/E_0$ tends to $\alpha_w/\alpha_0$ ($\leqslant 1$). For a given value of the exponent $\lambda$ and fixed values of $\alpha_0$ and $\alpha_w$ (= $1.26$), the relationship between $E/P$ and $\Phi_0$ (or between $E/E_0$ and $\Phi_0^{-1}$) can be obtained by solving numerically Eqs. (20) and (21). Similar calculations, more or less complicated, could be made with any Budyko function (Table 1). These results show that a Turc-Mezentsev curve (or any Budyko curve) generates a different curve when potential evaporation is given by $E_0$ instead of $E_p$. The new curve $B_1'$ is represented in Fig. 3a by comparison with the original one $B_1$ for two values of the shape parameter $\lambda$ ($0.5$ and $2$) and $b = 1$, assuming $\alpha_0 = \alpha_w = 1.26$. The new curve has a form similar to the original one, with the same limits at $0$ and $+\infty$, but it is higher or lower depending on the value of $\alpha_0$. In Fig. 3b the two curves are drawn when $\alpha_0$ is adjusted according to Eq. (22) to make same closer. It is

worthwhile noting also that $B_2'$ is different from $B_1'$, contrary to $B_2$ (Eq. 5) which is identical to $B_1$ (Eq. 4), but the two curves are very close, as shown in Fig. 4, and it is easy to verify they have the same value for $\Phi_0 = \Phi_0^{-1} = 1$.

We have now two sets of Budyko functions: $B_1'$ and $B_2'$ (Eqs. 20 and 21) involving $\Phi_0 = E_0/P$ and their corresponding original formulations $B_1$ and $B_2$ (Eqs. 4 and 5) as a function of $\Phi_p = E_p/P$. The question now is to find out the

value of $\alpha_0$ which allows $B_1'$ to be equivalent (or the closest) to the original Turc-Mezentsev function $B_1$. Both equations expressing $E/P$ as a function of an aridity index $\Phi$ ($\Phi_p$ or $\Phi_0$), the expression of $\alpha_0$ can be inferred by matching Eq. (20) and Eq. (4): for a given value of the aridity index $\Phi$, $B_1$ and $B_1'$ should give the same value of $E/P$. This leads to:

$$\alpha_0 = \frac{(1+b)\alpha_w}{1+b(1+\Phi^\lambda)^{-1/\lambda}} \ . \tag{22}$$

The same relationship (Eq. 22) is obtained by matching $B_2'$ with $B_2$. Putting the value of $\alpha_0$ defined by Eq. (22) into $B_1'$ and

$B_2'$ (Eqs. 20 and 21) leads to new transcendental equations linking $E/P$ and $\Phi_0$ (or $E/E_0$ and $\Phi_0^{-1}$) which are exactly equivalent to the original Turc-Mezentsev functions (Eqs. 4 and 5). Function $B_1'$ transforms into:

$$\frac{E}{P} + \left[\left(\frac{E}{P}\right)^{-\lambda} - 1\right]^{-1/\lambda} = \Phi_0 + \left(1 + \Phi_0^{-\lambda}\right)^{-1/\lambda} , \tag{23}$$

and $B_2'$ into:

$$\left\{1 + \left[1 + (\Phi_0^{-1})^{-\lambda}\right]^{-1/\lambda} - \frac{E}{E_0}\right\}^{-\lambda} = \left(\frac{E}{E_0}\right)^{-\lambda} - (\Phi_0^{-1})^{-\lambda} . \tag{24}$$

In the supplementary material (S3) we show that the original Turc-Mezentsev functions are the solutions of these transcendental equations. It is worthwhile noting also that when $\alpha_0$ is expressed by Eq. (22) and $\Phi_0$ tends to zero (or $\Phi_0^{-1}$ tends to infinite), $\alpha_w/\alpha_0$ in Eqs. (20) and (21) tends to $1$. This means that these equations have the same limits as their original equations (Eqs. 4 and 5).

For every value of $\lambda$ and $\Phi$, a unique value of $\alpha_0$ can be calculated by means of Eq. (22), $b$ and $\alpha_w$ being fixed. In

this equation $\alpha_0 = f(\lambda, \Phi)$, $\Phi$ represents climate aridity and $\lambda$ catchments characteristics in relation to its ability to evaporate (the greater $\lambda$, the higher its evaporation capability). The Priestley-Taylor coefficient $\alpha_0$ appears to be an increasing function of $\Phi$ and a decreasing function of $\lambda$. Fig. 5a shows the relationship between $\alpha_0$ and $\lambda$ for different values of $\Phi$. When $\lambda$ tends to zero (non-evaporative catchment), $\alpha_0$ tends to $(1+b)\alpha_w = 2\alpha_w$, whatever the value of $\Phi$. When $\lambda$ tends to infinity (i.e. very evaporating catchment), the limit of $\alpha_0$ depends on the value of $\Phi$: for $\Phi \leqslant 1$ the limit is $\alpha_w$ and for $\Phi > 1$ the limit is the

branch of the hyperbole $(1+b)\alpha_w\Phi/(b+\Phi) = 2\alpha_w\Phi/(1+\Phi)$. Fig. 5b shows the relationship between $\alpha_0$ and $\Phi$ for different values of $\lambda$. When $\Phi$ tends to $+\infty$ (very arid catchment), the coefficient $\alpha_0$ tends to $(1+b)\alpha_w = 2\alpha_w$. When $\Phi$ tends to $0$ (very humid catchment), $\alpha_0$ tends to $\alpha_w$. These results illustrate the simple functional relationship existing between the Priestley-Taylor coefficient, the Budyko shape parameter and the aridity index. Very similar results are obtained when the Fu-Zhang formulation is used instead of the Turc-Mezentsev one, as detailed in the supplementary material S4. In the last

supplementary material (S5), Figures 5a, b are redrawn with a value of $b = 4.5$, as obtained by Brutsaert (2015) from a

reformulated complementary relationship. The general shape of the curves is very similar, but the upper limits are much higher in agreement with a higher value of $b$.

## 5 Summary and conclusion

The Budyko curves have two different and equivalent dimensionless expressions: $B_1$ where $E/P$ is a function of the aridity index $\Phi_p = E_p/P$, and $B_2$ where $E/E_p$ is a function of $\Phi_p^{-1} = P/E_p$; any $B_1$ curve can be transformed into an equivalent $B_2$ curve and conversely. Among various Budyko type curves, the Turc-Mezentsev one (Eq. 4) with the shape parameter $\lambda$ was chosen because it is commonly used and has the remarkable property of having the same mathematical expression in both representations $B_1$ or $B_2$. Using Penman's equation (Eq. 2) to express potential evaporation and introducing the complementary evaporation relationship in the form of the Advection-Aridity model with its parameters $b$ and $\alpha_w$ (Eqs. 6 and

7), it was shown that the dimensionless drying power of the air $D = E_a/E_p$ expressed as a function of $\Phi_p^{-1}$ has upper and lower boundaries and that there is a functional correspondence between the Budyko and $D$ curves. Next, we examined the case where potential evaporation is expressed by the Priestley-Taylor equation ($E_0$ given by Eq. 3) with a varying coefficient $\alpha_0$ instead of the sounder Penman's equation. Introducing the CE relationship in the form of the Advection-Aridity model shows that the Turc-Mezentsev function linking $E/P$ to $\Phi_p = E_p/P$ (Eq. 4) transforms into a new transcendental form of the

Turc-Budyko function $B_1'$ linking $E/P$ to $\Phi_0 = E_0/P$ (Eq. 20), only numerically resolvable. The Priestley-Taylor coefficient $\alpha_0$ should have a specified value as a function of $b$, $\alpha_w$, $\lambda$ and $\Phi_0 = \Phi_p$ so that the two curves $B_1$ and $B_1'$ be equivalent. This means that the coefficient $\alpha_0$ [$\alpha_w \leqslant \alpha_0 \leqslant (1+b)\alpha_w$] is intrinsically linked to the shape parameter $\lambda$ of the Turc-Mezentsev function and to the aridity index.

Acknowledgements. The authors are very grateful to three anonymous reviewers and the Handling Editor for their constructive comments. They also thank the UMR LISAH for its scientific support and financial contribution.

## 6 List of symbols

| | |
|---|---|
| $B_1$ | function linking $E/P$ to $\Phi_p = E_p/P$. |
| $B_1'$ | function linking $E/P$ to $\Phi_0 = E_0/P$ given by Eq. (20). |
| 25   $B_2$ | function linking $E/E_P$ to $\Phi_p^{-1} = P/E_p$. |
| $B_2'$ | function linking $E/E_0$ to $\Phi_0^{-1} = P/E_0$ given by Eq. (21). |
| $b$ | asymmetry coefficient of the CE relationship (Eq. 6) |
| $D$ | function linking $E_a/E_p$ to $P/E_p$. |
| $D^*$ | difference between the upper and lower boundaries of $D$ [-]. |
| 30   $d^*$ | maximum difference between the Turc-Budyko curve and its upper limit [-]. |

$E$        actual evaporation [$LT^{-1}$].

$E_p$        potential evaporation expressed by Penman's equation [$LT^{-1}$].

$E_0$        potential evaporation expressed by the Priestley-Taylor equation [$LT^{-1}$].

$E_w$        wet environment evaporation (CE relationship) [$LT^{-1}$].

$P$        precipitation [$LT^{-1}$].

$E_a$        drying power of the air [$LT^{-1}$].

$E_{a,n1}$        lower limit of $E_a$ given by Eq. (12) [$LT^{-1}$].

$E_{a,n2}$        lower limit of $E_a$ given by Eq. (13) [$LT^{-1}$].

$E_{a,x}$        upper limit of $E_a$ given by Eq. (11) [$LT^{-1}$].

$R_n$        net radiation [$LT^{-1}$].

$\alpha_0$        varying coefficient of the Priestley-Taylor equation $E_0$ [-].

$\alpha_w$        $=1.26$: fixed coefficient of the Priestley-Taylor equation $E_w$ [-].

$\gamma$        psychrometric constant [$M L^{-1}T^{-2}\ °C^{-1}$].

$\varDelta$        slope of the saturated vapour pressure curve at air temperature [$M L^{-1}T^{-2}\ °C^{-1}$].

$\delta^*$        maximum difference between the $E_a/E_p$ curve and its lower boundary [-].

$\lambda$        shape parameter of the Turc-Mezentsev equation ($\lambda > 0$) [-].

$\varPhi_0$        aridity index calculated with $E_0$ ($\varPhi_0 = E_0/P$) [-].

$\varPhi_p$        aridity index calculated with $E_p$ ($\varPhi_p = E_p/P$) [-].

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

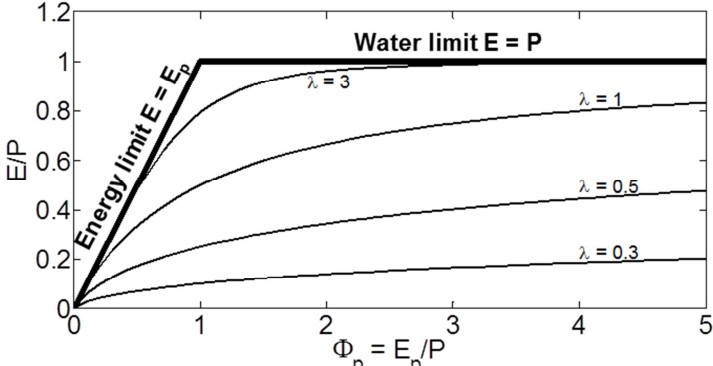

**Figure 1: The Turc-Mezentsev relationship Eq. (4) between the ratio *E/P* and the aridity index $\Phi_p = E_p/P$ for four values of the parameter $\lambda$ (*0.3, 0.5, 1* and *3*). The bold line indicates the upper limit of the feasible domain.**

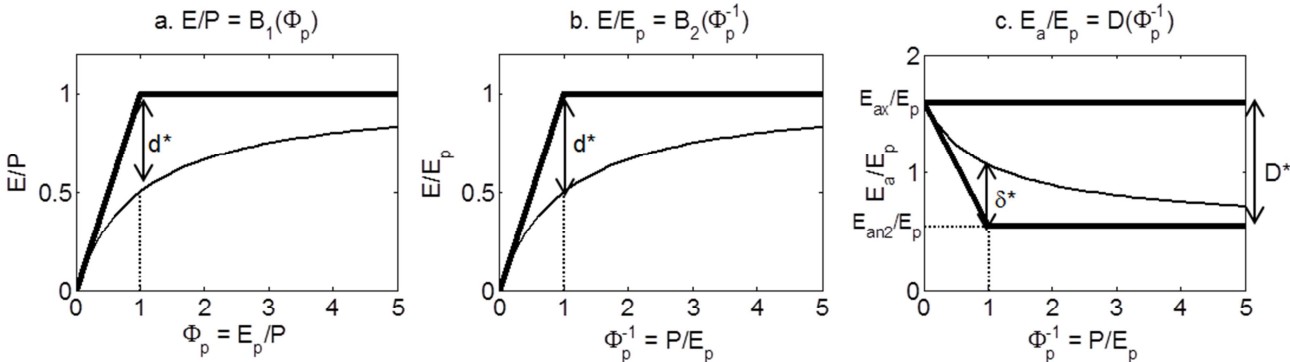

10     **Figure 2: Correspondence between the two forms of the Turc-Mezentsev functions ($E/P = B_1(\Phi_p)$ and $E/E_p = B_2(\Phi_p^{-1})$ given by Eqs. (4) and (5)) and the function defining the drying power of the air ($E_a/E_p = D(\Phi_p^{-1})$ given by Eq. (10)). The calculations are made with *b = 1*, $\lambda$ *= 1* and a temperature of *15°C*: *d\* = 0.50, D\* = 1.05* and $\delta$\* *= 0.52*. The bold lines indicate the upper limit of the feasible domain.**

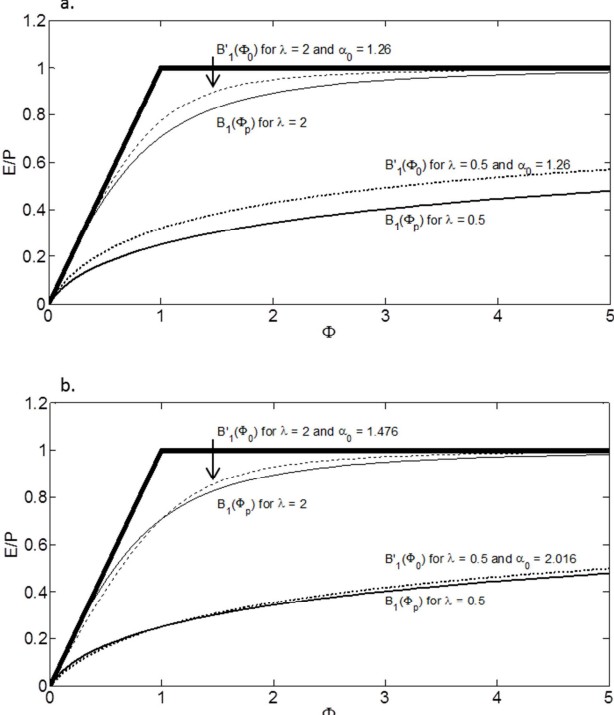

**Figure 3:** Comparison between the Turc-Mezentsev function $B_1(\Phi_p)$ (Eq. 4) in solid line and its corresponding function $B_1'(\Phi_0)$ (Eq. 20) in dotted line for two values of $\lambda$ (0.5 and 2) and $b =1$: (a) with $\alpha_0 = \alpha_w = 1.26$; (b) with $\alpha_0$ adjusted according to Eq. (22) for $\Phi = 1$. The x-axis legend $\Phi$ represents either $\Phi_p$ for $B_1(\Phi_p)$ or $\Phi_0$ for $B_1'(\Phi_0)$.

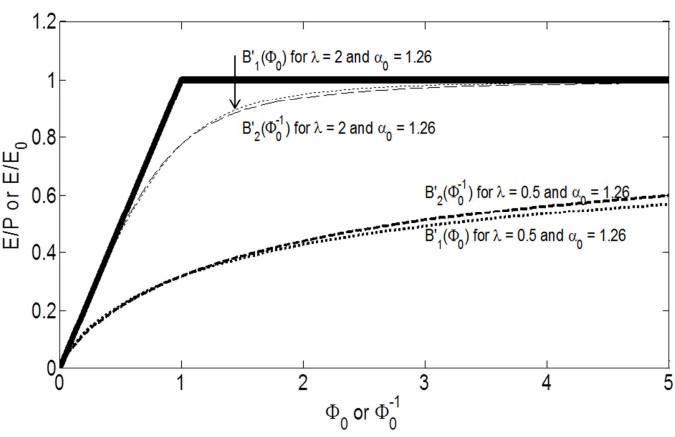

**Figure 4:** Comparison of functions $E/P = B_1'(\Phi_0)$ (Eq. 20) and $E/E_0 = B_2'(\Phi_0^{-1})$ (Eq. 21) for two different values of the shape
10   parameter $\lambda$ (0.5 and 2), $b =1$ and $\alpha_0 = 1.26$.

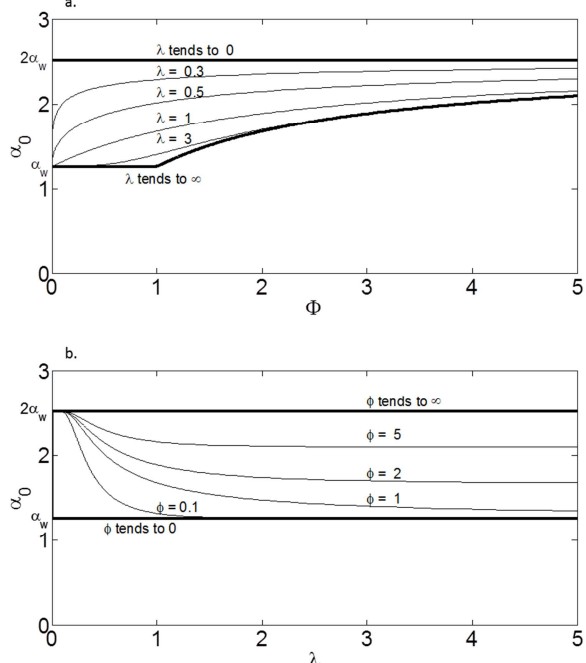

**Figure 5: Variation of the Priestley-Taylor coefficient $\alpha_0$ (Eq. (22) with $b = 1$ and $\alpha_w = 1.26$): (a) as a function of the aridity index $\Phi$ for different values of the shape parameter $\lambda$ of the Turc-Mezentsev function; (b) as a function of $\lambda$ for different values of the aridity index $\Phi$. The bold lines indicate the upper and lower limits of the feasible domain.**

**Table 1: Different expressions of the Budyko functions as a function of the aridity index $\Phi_p$.**

| Equation | Reference |
|---|---|
| $E/P = \left\{ \Phi_p \tanh(\frac{1}{\Phi_p})[1 - \exp(-\Phi_p)] \right\}^{1/2}$ | Budyko (1974) |
| $E/P = \Phi_p \left[ 1 + (\Phi_p)^{\lambda} \right]^{-\frac{1}{\lambda}}$ | Turc (1954) with $\lambda = 2$, Mezentsev (1955), Yang et al. (2008) |
| $E/P = 1 + \Phi_p - \left[ 1 + (\Phi_p)^{\omega} \right]^{\frac{1}{\omega}}$ | Fu (1981), Zhang et al. (2004) |
| $E/P = \dfrac{1 + w\Phi_p}{1 + w\Phi_p + \Phi_p^{-1}}$ | Zhang et al. (2001) |
| $E/P = \Phi_p \left( \dfrac{k}{1 + k\Phi_p^{\,n}} \right)^{1/n}$ | Zhou et al. (2015) |

