# Peer review of "Matching the Budyko functions with the complementary evaporation relationship: consequences for the drying power of the air and the Priestley-Taylor coefficient"

_Hydrology and Earth System Sciences, 2016_

## Referee Comment (RC1) · Anonymous Referee #1 · 25 May 2016

The manuscript by Lhomme and Moussa addresses the interesting link between the complementary relationship and Budyko curve. I found the paper interesting and well written. I ahve a few comments: - Actually the first Budyko curve was in terms of net radiation and not potential evaporation. This should be mentioned because there was no drying power in the original framework! - p2 line 29-31: I would remove this sentence on potential evaporation because it is inconsistent with the assumption that it is used for the wet evaporation - p3 line 2 see also Lintener et al. 2015 for an analytical expression of alpha (this shoudl in fact similarities or differences with this expression should be discussed in the context of the retrieved alpha as a function of the shape parameter

of the Budyko curve - at least qualitatively) - line 8: should mention Szilagyi, J., and J. Jozsa (2009), Complementary relationship of evaporation and the mean annual water-energy balance, Water resources Research, 45(9), doi:10.1029/2009WR008129. - line 17 p4: we know this is not correct b>1, please comment or add. - reformulate line 26: rather "as a consequence of land-atmosphere interactions " ...." as expressed by the CR"

---

## Referee Comment (RC2) · Anonymous Referee #2 · 12 Jun 2016

Title: Matching the Turc-Budyko functions with the complementary evaporation relationship: consequences for the drying power of the air and the Priestley-Taylor coefficient
Authors: Lhomme and Moussa

This manuscript aimed to relate the Budyko curve with the complementary relation of evaporation and then explore the varying coefficient alfa in the Priestley-Taylor equation (which was redefined) to calculate potential evaporation. Furthermore, a function was proposed to relate this coefficient with the shape parameter of the Budyko hypothesis and aridity index. It is an interesting research. However, some improvements are required. Especially the significance needs further highlighting.

Detailed comments

1.  In this manuscript, the authors introduced a new parameter $\alpha_0$ into the complementary relationship between potential evaporation and actual evaporation. In fact, E0 estimated by equation (3) and Ep estimated by equation (2) are equivalent in this manuscript. Therefore, $\alpha_0$ represents the ratio between radiative item and aerodynamic item in the potential evaporation calculated by the Penman equation. The variation in $\alpha_0$ can be revealed according to Penman equation. Therefore, more discussion was required to show the theoretical significance of this manuscript. In application of estimating actual evaporation, this method has a precondition, which is to determine $\alpha_0$ according to Budyko curve. However, the Budyko curve has an ability of estimating actual evaporation. What is the objective of estimating $\alpha_0$ using the Budyko curve and then estimating actual evaporation using the CE?

2.  According to equations (6), (7) and (3) (If E0 and Ep are equivalent), it can yield

$$E = \left(2\alpha_w - \alpha_0\right)\frac{\Delta}{\Delta+\gamma}R_n .$$

Where $\alpha_w = 1.26$, $\alpha_0$ is determined by aridity index and the parameter $\lambda$, which is a constant in a special catchment because of constant aridity index and the parameter $\lambda$. Therefore, E only depends on Rn (temperature has a small impact on $\Delta$ and $\gamma$). The rationality needs more discussion.

3.  In this manuscript, $\alpha_0$ was named the Priestley-Taylor coefficient to calculate potential evaporation, and at the same time, another Priestley-Taylor coefficient $\alpha_w = 1.26$ in the Priestley-Taylor equation was used to calculate the wet environment evaporation. It is likely to cause confusion.

4. The timescale should be pointed out when relate the BT to CE, because the BT is general used on the long-term time scale or annual scale.
5. Turc-Budyko curves should be replaced with Budyko-Type curves.
6. P.4, line 24, more explanation on $\alpha_w \leqslant \alpha_0 \leqslant 2\alpha_w$ are required.

---

## Referee Comment (RC3) · Anonymous Referee #3 · 24 Jun 2016

This paper aims to relate the complementary relationship and the Budyko framework. It is very interesting, and may be useful for the understanding of evaporation estimation. However, some improvements are required. I am having some comments below.

1. Different definitions of "potential evaporation" need distinguishing. First: in the Budyko framework, " potential evaporation " is defined as energy supply for evaporation, which is estimated by solar radiation, Penman equation, or Priestley-Taylor equation. They were used in same equations without distinguishing their differences. So, the question is, why Penman evaporation is used in Eq. (1), and denoting Priestley-

Taylor evaporation indirectly through the complementary relationship?

2 Why using Priestley-Taylor equation by Eq. (3) and (7)? What is the difference? Please give more explanations.

3. What is the purpose or significance of the work? Improving evaporation estimation? Help to determine the Priestley-Taylor coefficient?

4. Some generalized complementary relationship (Brutsaert, 2015, Han et al., 2012) were proposed in recent publications. However, the advection-aridity model of Brutsaert (1979) is used to denote the complementary relationship model in this paper. As a result, the linking proposed in this paper may be not generalized.

5. In section 3, the drying power of the air is used, and the psychrometric constant and the slope of the saturated vapor pressure curve at air temperature have to be taken as variables. If using the aerodynamic term instead, the relationship may be more clear.

---

## Author Response (AR1)

**Responses to reviewers**

We are very grateful to three anonymous reviewers and the editor for their constructive comments of the manuscript. All the changes or adding made to the submitted MS are in red in the new version. Note that the term "Turc-Budyko function" used in the submitted MS was replaced by "Budyko function" for more coherence with the literature. The abstract was also modified to better highlight the significance of the work.

**Editor comments to the authors**

*1) Highlight the significance of the work;*

Additional comments (in red) have been added in the introduction and in section 2 to make clearer the significance of the work.

*2) Recalculate based on the generalised CR model (Brutsaert, 2015, Han et al., 2012), as suggested by the Referee #3;*

As detailed in sections #1.5 and #3.4 below, new calculations have been made with a generalized form of the CR model including a variable coefficient $b$ instead of the fixed value $b = 1$ (Eq. 6).

*3) P.2, line 1, "and soil water storage (Li et al., 2013; Yang et al., 2007)." ---> 'and soil water storage (Li et al., 2013; Yang et al., 2007), catchment slope (Yang et al., 2014)"*
*Yang, H., Qi, J., Xu, X., Yang, D., and Lv, H.: The regional variation in climate elasticity and climate contribution to runoff across China, Journal of Hydrology, 517, 607–616, 2014.*

This new reference (influence of catchment slope on the shape parameter of the Budyko function) has been added P2 line 1.

**Referee #1**

1.  *Actually the first Budyko curve was in terms of net radiation and not potential evaporation. This should be mentioned because there was no drying power in the original framework!*

It is true that there was no drying power in the original framework and in some subsequent works, e.g. Choudhury (1999), Donohue et al. (2007). Consequently, a comment was added to recognize that (P2 line 27).

> 2. *p2 line 29-31: I would remove this sentence on potential evaporation because it is inconsistent with the assumption that it is used for the wet evaporation.*

Some additional comments (P5 lines 6-9) were made to clearly specify which is the original Priestley-Taylor equation with a fixed coefficient $\alpha_w = 1.26$ (used to calculate wet environment evaporation $E_w$ in the AA model) and which is the Priestley-Taylor type equation (used to estimate potential evaporation $E_0$) with a variable coefficient $\alpha_0$.

> 3. *p3 line 2 see also Lintner et al. 2015 for an analytical expression of alpha (in fact similarities or differences with this expression should be discussed in the context of the retrieved alpha as a function of the shape parameter of the Budyko curve - at least qualitatively).*

In fact, as far as we understand, the analytical expression of alpha in Lintner et al. (2015, Eq. 13) applies to $\alpha_w$ which defines the wet environment evaporation $E_w$ and set to 1.26 in the AA model used in our analysis. Our analytical expression of alpha (Eq. 22) applies to $\alpha_0$ (which defines potential evaporation $E_0$) and not to $\alpha_w$. A comment has been added P5 line 2.

> 4. line 8: should mention Szilagyi, J., and J. Jozsa (2009), Complementary relationship of evaporation and the mean annual water energy balance, Water Resources Research, 45(9), doi:10.1029/2009WR008129.

This reference is relevant and was added (P3 line 15).

> 5. *line17 p4: we know this is not correct b>1, please comment or add.*

We made new calculations with a not-fixed value of $b$. It is not very complicated, in fact. In the revised paper, all the equations are modified by including the parameter $b$. The value of $b$ is discussed at the end of the paper in the light of the recent paper of Brutsaert (2015). His generalized form of the complementary relationship suggests that $b= 4.5$ would be more appropriate to account for the asymmetry of the CE relationship. The figures of the main text are kept with $b = 1$ (original AA model), but they are redrawn with $b = 4.5$ in the supplementary materials (S.5).

> 6. *reformulate line 26: rather "as a consequence of land-atmosphere interactions " ...." as expressed by the CR".*

The referee is right. This part of the phrase was changed (P5 line 18).

**Referee#2**

1. *In this manuscript, the authors introduced a new parameter $\alpha_0$ into the complementary relationship between potential evaporation and actual evaporation. In fact, $E_0$ estimated by equation (3) and Ep estimated by equation (2) are equivalent in this manuscript. Therefore, $\alpha_0$ represents the ratio between radiative item and aerodynamic item in the potential evaporation calculated by the Penman equation. The variation in $\alpha_0$ can be revealed according to Penman equation. Therefore, more discussion was required to show the theoretical significance of this manuscript. In application of estimating actual evaporation, this method has a precondition, which is to determine $\alpha_0$ according to Budyko curve. However, the Budyko curve has an ability of estimating actual evaporation. What is the objective of estimating $\alpha_0$ using the Budyko curve and then estimating actual evaporation using the CE?*

Maybe the text was not sufficiently explicit and clear, but our objective is not about estimating actual evaporation, or at least it is not our main concern. Having defined the Priestley-Taylor coefficient $\alpha_0$ in the way of Eq. (3), as a means to estimate potential evaporation *Ep*, we simply show there is a functional relationship between this coefficient $\alpha_0$ and the shape parameter $\lambda$ of the Budyko curve, this relationship being a direct consequence of the CE relationship. This point is made clearer in the new manuscript (P3 lines 10-13).

2. *According to equations (6), (7) and (3) (If $E_0$ and Ep are equivalent), it can yield*

$$E = (2\alpha_w - \alpha_0) \frac{\Delta}{\Delta + \gamma} R_n$$

*Where $\alpha_w = 1.26$, $\alpha_0$ is determined by aridity index and the parameter $\lambda$, which is a constant in a special catchment because of constant aridity index and the parameter $\lambda$. Therefore, E only depends on Rn (temperature has a small impact on $\Delta$ and $\gamma$). The rationality needs more discussion.*

The equation is correct, but we cannot say that *E* only depends on $R_n$, since $\alpha_0$ is a function of $\lambda$ and of the aridity index $\Phi$. We can simply say that in a given catchment characterized by fixed values of $\lambda$ and $\Phi$, *E* depends on $R_n$ and on $\lambda$ and $\Phi$ trough $\alpha_0$.

3. *In this manuscript, $\alpha_0$ was named the Priestley-Taylor coefficient to calculate potential evaporation, and at the same time, another Priestley-Taylor coefficient $\alpha_w = 1.26$ in the Priestley-Taylor equation was used to calculate the wet environment evaporation. It is likely to cause confusion.*

It is the point which should be made clearer. In fact, in our analysis two Priestley-Taylor coefficients are defined in relation to the CE relationship: one ($\alpha_w$) is used to define the wet environment evaporation $E_w$ and the other ($\alpha_0$) serves to calculate the potential evaporation $E_0$, which is a substitute for $E_p$. This point is explained P5 lines 6-9.

4. *The timescale should be pointed out when relate the BT to CE, because the BT is general used on the long-term time scale or annual scale.*

It is true that the Budyko curves were initially derived and used on long time scales, but they have been downscaled to the season or the month by some authors (Zhang et al., 2008; Du et

al., 2016; Greve et al., 2016). As pointed out by Lintner et al (2015, p2120), observational data confirm that the CE relationship holds on daily to annual timescales. Some comments are added in the revised manuscript P5 lines 9-12.

5. *Turc-Budyko curves should be replaced with Budyko-Type curves.* OK

6. *P.4, line 24, more explanation on $\alpha_w \leqslant \alpha_0 \leqslant 2\alpha_w$ are required.*

This is a direct consequence of the CE relationship (Eq. 6 with $b = 1$) replacing $E_p$ by $E_0$. Additional explanation was added P5 line 13.

**Referee #3**

1. *Different definitions of "potential evaporation" need distinguishing. First: in the Budyko framework, "potential evaporation" is defined as energy supply for evaporation, which is estimated by solar radiation, Penman equation, or Priestley-Taylor equation. They were used in same equations without distinguishing their differences. So, the question is, why Penman evaporation is used in Eq. (1), and denoting Priestley-Taylor evaporation indirectly through the complementary relationship?*

In fact, when Penman's equation is used to estimate potential evaporation $E_p$ simultaneously in the CE relationship and in the Budyko function, the question does not exist. It is when $E_0$ (Priestley-Taylor equation with a given coefficient $\alpha_0$) is used instead of $E_p$ (Penman), that the problem arises and our analysis becomes relevant.

2. *Why using Priestley-Taylor equation by Eq. (3) and (7)? What is the difference? Please give more explanations.*

The CE relationship involves two kinds of "potential" evaporation, a "true" potential evaporation represented by Penman equation ($E_p$) and estimated by $E_0$ ($\alpha_0$) and a wet environment "potential" evaporation estimated by $E_w$ ($\alpha_w$). Both $E_0$ and $E_w$ are estimated via the same form of the Priestley-Taylor equation, but with different coefficients ($\alpha_0$ and $\alpha_w$). This clarification was added in the new manuscript P5 lines 6-9.

3. *What is the purpose or significance of the work? Improving evaporation estimation? Help to determine the Priestley-Taylor coefficient?*

As previously discussed, our main purpose is not improving evaporation estimation, or maybe indirectly. It is determining the Priestley-Taylor coefficient $\alpha_0$ (the one expressing potential evaporation $E_0$) as a function of the parameters defining the Budyko function ($\lambda$ and $\Phi$). This point is made clearer P3 lines 9-13.

4. *Some generalized complementary relationships (Brutsaert, 2015, Han et al., 2012) were proposed in recent publications. However, the advection-aridity model of Brutsaert (1979) is used to denote the complementary relationship model in this paper. As a result, the linking proposed in this paper may be not generalized.*

As already said in our response #1.5, we made new calculations with a not-fixed value of $b$ in the complementary relationship (it is relatively simple). Consequently the new linking between $\alpha_0$ and $\lambda$ (Eq. 22) is based on a more general form of the complementary relationship.

5. *In section 3, the drying power of the air is used, and the psychrometric constant and the slope of the saturated vapor pressure curve at air temperature have to be taken as variables. If using the aerodynamic term instead, the relationship may be more clear.*

The relationship would be certainly a little bit clearer. However, temperature has a relatively small impact on $\gamma$ and $\Delta$. And more importantly, $E_a$ has a physical significance per se (equivalent to $R_n$ in the Penman equation), which is not the case for the aerodynamic term. It is the reason why we prefer to keep $E_a$.

**References**

Brutsaert, W.: A generalized complementary principle with physical constraints for land-surface evaporation, Water Resources Research doi:10.1002/2015WR017720, 2015.

Donohue, R.J., Roderick, M.L., McVicar, T.R.: On the importance of including vegetation dynamics in Budyko's hydrological model, Hydrol. Earth Syst. Sci., 11, 983-995, 2007.

Du, C., Sun, F., Yu, J., Liu, X., Chen, Y.: New interpretation of the role of water balance in an extended Budyko hypothesis in arid regions, Hydrol. Earth Syst. Sci., 20, 393-409, 2016

Greve, P., Gudmundsson, L., Orlowsky, B., Seneviratne, S.I.: A two-parameter Budyko function to represent conditions under which evapotranspiration exceeds precipitation, Hydrol. Earth Syst. Sci., 20, 2195-2205, 2016.

Zhang, L., Potter, N., Hickel, K., Zhang, Y., Shao, Q.: Water balance modeling over variable time scales based on the Budyko framework - Model development and testing, Journal of Hydrology 360, 117-131, 2008.

---

## Referee Report (RR1)

Comment on "Matching the Budyko functions with the complementary evaporation relationship: consequences for the drying power of the air and the Priestley-Taylor coefficient" by Jean-Paul Lhomme, Roger Moussa

This paper aims to relate the complementary relationship and the Budyko framework. As the authors have responded the comments well, I think the manuscript should be accepted for publication. However, there is one typo:

1. Page 1, Line 28: "Wang et al., 20015" should be "Wang et al., 2015"

---

## Author Response (AR2)

**Responses to the reviewers and the editor**

We thank the two anonymous reviewers and the editor for their constructive comments of the manuscript.

**Editor**

*I would like to suggest a minor revision of this work based on the comments raised by the reviewers and on my own judgement. Please pay special attentions to: 1) re-clarify the statement on the timescale of the Budyko and complementary relationship, as suggested by the Referee #3. Although the Budyko curve has been applied for the season or month, the equation used in the manuscript (with precipitation as water availability) is valid only at a long-time scale; 2) revise the typos.*

See responses below.

**Reviewer 1**

*This paper aims to relate the complementary relationship and the Budyko framework. As the authors have responded the comments well, I think the manuscript should be accepted for publication. However, there is one typo:*
*1. Page 1, Line 28: "Wang et al., 20015" should be "Wang et al., 2015"*

The typo was corrected.

**Reviewer 2**

*The authors did a good revision on the original manuscript following the suggestions and comments from editor and reviewers. However, I don't think the statement on the timescale of the Budyko and complementary relationship is valid, at least in this manuscript, because: (1) line 5-10, equation (1) from the Budyko equation is valid only at a long-time scale; (2) equation (5) is valid only at a long-time scale also, which is the basis of equation (10). To be sure, the complementary relationship is valid at different time scales.*

We recognize that our statement on timescale was not perfectly clear in the previous manuscript and certainly put at the wrong place (P5 lines 9-12). Consequently the statement was removed from its original place and split into two new statements in the introduction. A first statement (P2 lines 7-9) recalls that the Budyko functions initially established for long-time scales have been downscaled to the season or the month through analytical adjustments. The second statement (P2 lines 26-28) confirms that the CE relationship holds on different timescales from the day to the year, which justifies the matching between the two relationships (CE and Budyko) on long timescales. A new reference was added (Morton, 1983) to support the long timescale approach for the CE relationship.